# Slip Activation Potential of Fractures in the Crystalline Basement Rocks of Kuujjuaq (Nunavik, Canada) to Assess Enhanced Geothermal Systems Development

**Mafalda M. Miranda [1,*], Ali Yaghoubi [2,3], Jasmin Raymond [1], Andrew Wigston [3] and Maurice B. Dusseault [2]**

[1] INRS—Institut National de la Recherche Scientifique, 490 rue de la Couronne, Quebec, QC G1K 9A9, Canada; jasmin.raymond@inrs.ca

[2] Department of Earth & Environmental Sciences, Centre for Environmental and Information Technology (EIT), 200 University Ave. W, Waterloo, ON N2L 3G1, Canada; ali.yaghoubi@uwaterloo.ca (A.Y.); mauriced@uwaterloo.ca (M.B.D.)

[3] Natural Resources Canada, 1 Haanel Drive, Ottawa, ON K1A 1M1, Canada; andrew.wigston@nrcan-rncan.gc.ca

\* Correspondence: mafalda_alexandra.miranda@inrs.ca

**Abstract:** This work presents an estimate of the slip activation potential of existing fractures in a remote northern community located on Canadian Shield rocks for geothermal purposes. To accomplish this objective, we analyzed outcrop analogues and recorded geometrical properties of fractures, namely the strike and dip. Then, we estimated the stress regime in the study area through an empirical approach and performed a probabilistic slip tendency analysis. This allowed us to determine the slip probability of the pre-existing fractures at the current state of stress, the orientation of fractures that are most likely to be activated and the fluid pressures needed for the slip activation of pre-existing fractures, which are key aspects for developing Enhanced Geothermal Systems. The results of this simple, yet effective, analysis suggest that at the current state of stress, the pre-existing natural fractures are relatively stable, and an injection pressure of about 12.5 MPa/km could be required to activate the most optimally oriented fractures to slip. An injection of water at this pressure gradient could open the optimally oriented pre-existing fractures and enhance the permeability of the reservoir for geothermal fluid extraction. The information described in this paper provides a significant contribution to the geothermal research underway in remote northern communities.

**Keywords:** Monte Carlo analysis; scanline sampling; fracture network; empirical stress regime; geothermal energy; enhanced geothermal systems; Canadian Shield

## 1. Introduction

Canada is an energy-intensive and energetically contrasting country (Figure 1; [1]). Canadians in the densely populated southern region are connected to a provincially interconnected electricity grid, powered by a variety of energy sources. In the Northwest Territories and the Yukon, most communities are connected to independent territorial grids dominated by hydro power. The situation in Nunavut and Nunavik (an administrative region of Quebec) is however quite different as there are no territorial grids, and communities generate their own electricity using generators connected to micro-grids [1,2]. Despite some renewable power projects and several initiatives to diversify the Nunavut and Nunavik communities' energy portfolio, diesel is still their main source of energy [1–3].

Beyond being not connected to the provincial power grid, the communities in Nunavut and Nunavik are also not connected by road. Airplanes and boats are the only means of transportation for goods, materials, and people to reach the communities. The residents of Nunavut and Nunavik are thus compelled to provide for themselves, since the communities cannot share infrastructural assets and resources.

Another particularity of the communities in Nunavut and Nunavik is that they are located on the crystalline basement rocks of the Canadian Shield. Metamorphic and igneous rocks, Archean to Paleoproterozoic in age, dominate the geology of these regions [4,5].

An assessment of the Canadian geothermal potential carried out by Grasby et al. [6] suggested that engineered/enhanced geothermal systems (EGS) could be an option to extract the geothermal resource in the communities located on the Canadian Shield (Figure 1).

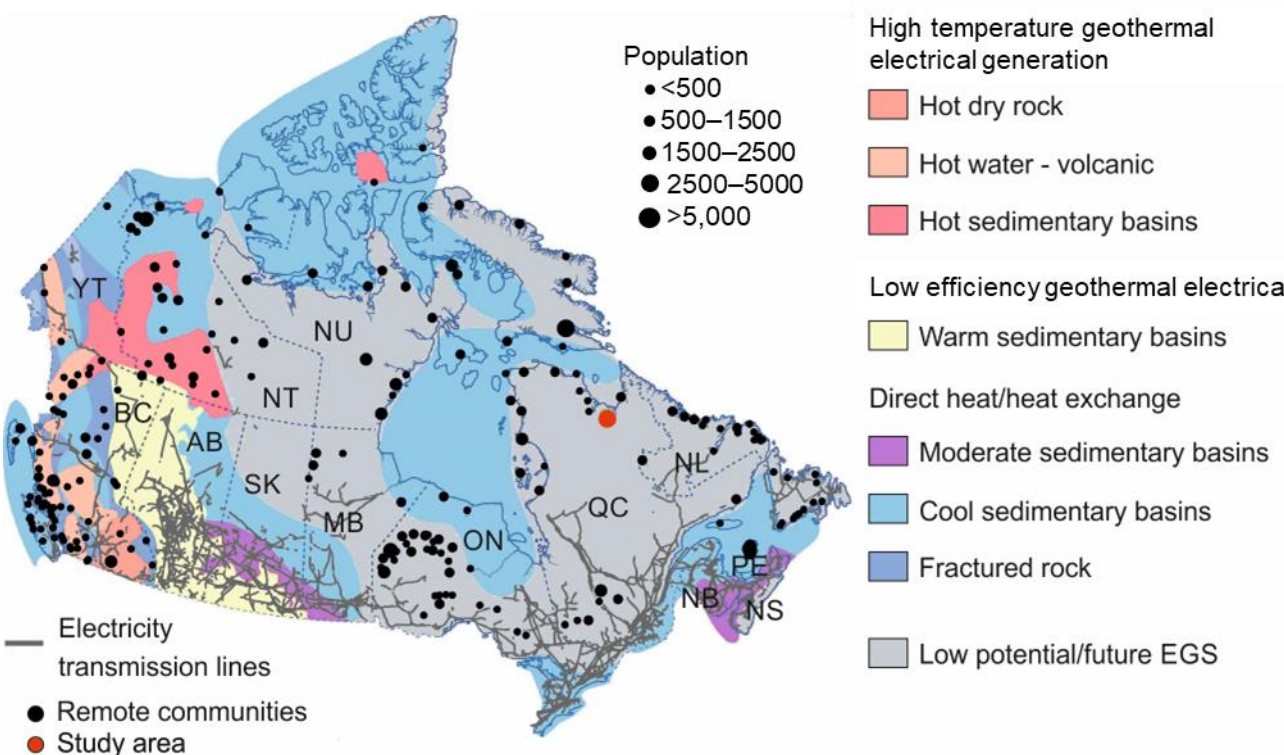

**Figure 1.** Distribution of geothermal potential in Canada by end use and location of power grid and remote communities (adapted from [6,7]). Provinces: BC—British Columbia; AB—Alberta; SK—Saskatchewan; MB—Manitoba; ON—Ontario; QC—Quebec; NL—Newfoundland and Labrador; NB—New Brunswick; NS—Nova Scotia; PE—Prince Edward. Territories: YT—Yukon Territory; NT—Northwest Territories; NU—Nunavut.

EGS is an emerging technology that arose from a concept initiated in Los Alamos (USA), Cornwall (UK), and Soultz-sous-Forêts (France) for exploiting geothermal resources in low-permeability rocks [8–14]. The central concept is to engineer fluid flow pathways between two or more wells to permit circulation of fluid through the reservoir rock mass to extract heat at rates of commercial interest.

EGS technology is needed in low-porosity fractured crystalline rocks whose natural permeability is very low due to poor hydraulic connectivity within the natural fracture network, requiring stimulation to increase the permeability of the rock mass [12]. Generally, hydraulic stimulation techniques are used for this objective.

During hydraulic stimulation, a high-pressure fluid is injected through the wellbore into the crystalline rock mass, leading to the shearing and opening of natural fractures, enhancing the hydraulic connectivity of these pre-existing fractures in the reservoir far field [11–13,15,16], and perhaps developing new fractures and extending existing fractures. In a successful hydraulic stimulation treatment, geothermal wells become better connected to the reservoir far field, and the geothermal resource can be exploited by circulating fluid between wells through the newly stimulated reservoir where the newly conductive fractures act as natural heat exchangers.

Permeability enhancement via hydraulic stimulation is commonly associated with two main mechanisms [15]: (1) hydraulic fracturing (opening of existing fractures and initiation and propagation of new tensile fractures), and (2) hydraulic shearing (activation of existing discontinuities in shear, leading to irreversible dilation). These two mechanisms are often referred to as Mode I and Mode II fracturing and correspond to the "end members" of the stimulation mechanisms while mixed-mode stimulation mechanisms can also occur [17,18]. This is because the pressure required for natural fractures to fail in shear may be lower than the pressure required for pure tensile fracturing (Figure 2; [19]).

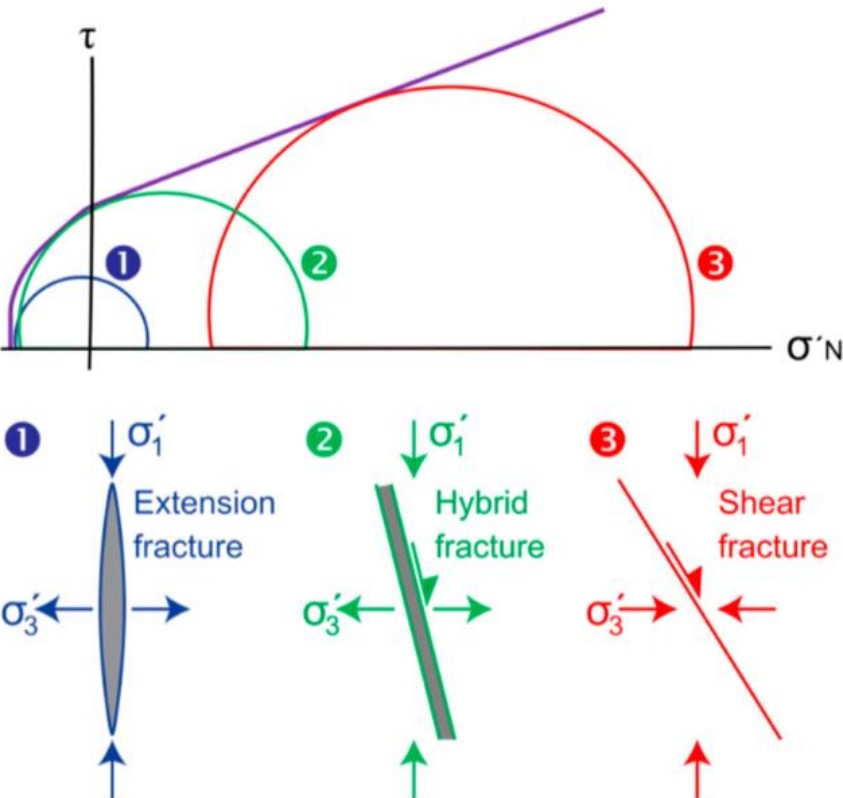

**Figure 2.** Example of Mohr diagram showing the fields in which extension (1), hybrid (2) and shear (3) fractures occur [20]. Reprinted/adapted with permission from Ref. [20]. Copyright 2021, The Authors.

Fluid injection changes the in situ pore pressure, and the effective normal stresses on the relatively weaker slip planes are reduced, diminishing the frictional strength. This leads to the activation of slip along faults and joints within the fracture network that are favorably aligned with the principal stress orientations (i.e., at a $30° \pm 5°$ from the maximum horizontal stress [21]). Shearing and dilation of natural pre-existing fractures result in a permanent increase in the rock mass permeability to levels suitable for the extraction of geothermal resources [12,13,15,16]. A review of hydraulic stimulation experiments highlights that hydroshearing will be more effective in areas with high differential stresses and hence high shear stress (Figure 2; [22]), a condition referred to as "critically stressed" if the joint is close to the onset of the shear slip.

However, the activation of natural fractures can generate seismic events. In fact, a major issue associated with hydraulic stimulation is induced seismicity, especially in critically stressed crystalline basement rocks [23,24].

Monitoring of injection-induced seismicity is commonly used to assess how successful hydraulic stimulation treatments were and to evaluate the development of flow pathways within the reservoir. The microseismic event cloud collected as part of the monitoring program helps in designing further drilling into the reservoir [13,25].

Although induced seismicity is necessary for reservoir development, if not properly assessed, predicted, and controlled, it can become a major barrier to the development of geothermal energy [26]. Most induced events are of low magnitude (below local magnitude $M_L$ = 2) and produce ground motions far below the threshold to be felt [27]. However, large magnitude ($M_L$ > 2.5) events may be felt at the surface and lead to the suspension of an EGS project [27], such as occurred with the Basel Deep Heat Mining project in Switzerland [22,26].

A primary induced seismicity assessment tool is to test the likelihood of fracture activation (i.e., shear slip) related to stress field perturbations [28]. This can be carried out by applying the technique of slip tendency analysis [29] and Mohr diagrams to answer fundamental questions that arise at early stages of geothermal exploration [20,30]:

1.  What is the probability of shear slip on pre-existing fractures at the current state of stress?
2.  Which orientations of fractures are most likely to be activated?
3.  What in situ fluid pressure is required to overcome the shear stress and activate pre-existing fractures?

The propensity of a surface to undergo shear slip depends on its frictional characteristics and the ratio of shear to normal effective stress acting on the surface—the slip tendency [29]. Slip tendency analysis allows us to investigate slip potential along any fracture orientation with respect to the ambient stress field, and consequently, assess the slip probability subject to a pore pressure perturbation.

Usually, a slip tendency analysis is carried out in a deterministic manner, considering a single analysis as definitive. However, subsurface conditions are uncertain and each geomechanical parameter that influences the rate and magnitude of injection-induced seismicity has inherent levels of uncertainty. Understanding this uncertainty is helpful to make well-informed decisions regarding user-controlled parameters (e.g., injection pressure, flow rate, fluid volume, etc.) [31]. Thus, a probabilistic slip tendency analysis is a relevant risk management and mitigation tool [31].

A probabilistic slip tendency analysis considers the inherent uncertainties associated with each input variable, including stress magnitudes and orientations, fault dip directions, angles, and frictional strengths. In a probabilistic slip tendency analysis, the uncertainties of the input variables (e.g., pressure, distance) are considered, allowing for a more comprehensive evaluation of slip probability in various scenarios. The slip tendency is directly influenced by the magnitude of the principal stresses, meaning that any changes in these stresses will directly impact the outcome. Therefore, a probabilistic approach provides a more thorough and suitable method for assessing the slip probability across multiple scenarios.

Minimum requirements to carry out a slip tendency analysis, either deterministic or probabilistic, are an accurate knowledge of the stress field and of the geometrical properties of fractures, namely the strike and dip.

In the absence of subsurface data, outcrop analogues are useful tools to acquire information regarding the geometrical properties of fractures [32–35]. Care is necessary when analyzing fractures from exposed outcrops, since these may be a result of exhumation and stress relief and may not persist in the deep subsurface [32,35]. Nevertheless, outcrop analogues are particularly useful in remote environments where subsurface data are unavailable, and it can be assumed that for a first-order analysis in crystalline rock masses, there are no significant differences between rocks at the surface and at depth.

The International Society for Rock Mechanics (ISRM) suggests overcoring and hydraulic fracturing and/or hydraulic testing of pre-existing fractures as methods to estimate rock stress [36–38]. However, in remote areas, these methods often cannot be applied, and a first-order estimate of the stress field relies on a compilation of available structural and rock mass fabric data and the application of empirical relationships. This is the first step to establish an in situ stress model [39].

The objective of this work is to estimate the slip activation potential of existing fractures in subsurface crystalline basement rocks from outcrop analogues and empirical stress predictions. The interactive computer tool developed by Yaghoubi et al. [31,40] for a probabilistic assessment of the slip tendency of faults at the Alberta No. 1 geothermal project site in Alberta, Canada, and in the Montney Formation, is used to make the necessary calculations and to plot Mohr diagrams to analyze the relationships between stresses, fluid pressure, and fractures, and provide a graphic representation of the effective stress states and slip potential.

First, we present the results of the fracture network characterization based on outcrop analogues, and the estimates of the stress field based on data compilation and empirical relationships. This information is then used in the slip tendency analysis. The remote community of Kuujjuaq in Nunavik (study area; Figure 1) is used as an example. This community was selected since geothermal research has been previously conducted at this location [41–50], and this work represents a further contribution.

## 2. Fracture Network Characterization

Geometrical properties of fractures (i.e., strike and dip) were sampled on exposed outcrop surfaces using the scanline sampling method (Figure 3) and a transit compass corrected for the $-21°1'$ magnetic declination in Kuujjuaq [51]. Since no distinction between joints and faults was made while collecting information, the term fracture is preferred and used in this work to group joints and faults.

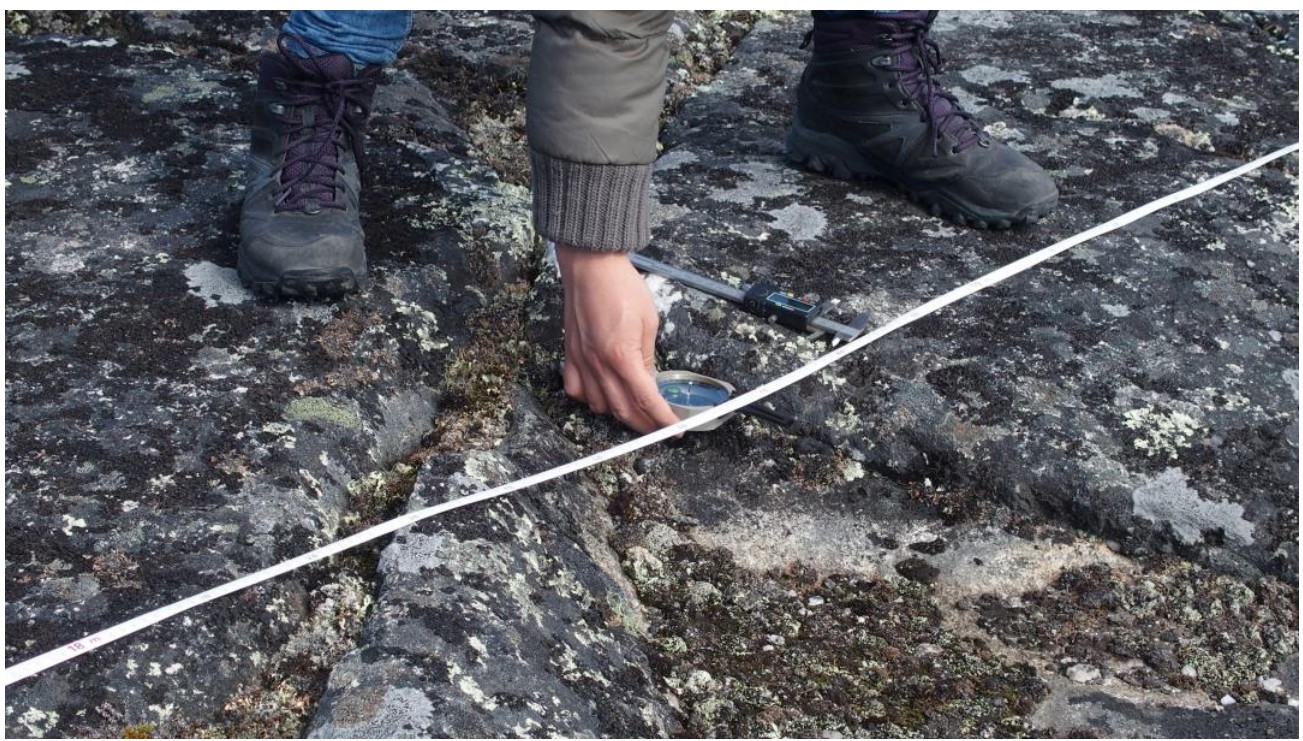

**Figure 3.** Example of scanline sampling method on an exposed outcrop surface in the study area.

In this method, a tape is laid on the outcrop surface perpendicular to the fracture sets observed [52,53]. Information about the strike and dip of each fracture intersecting the tape is then recorded.

Six fracture sampling areas around the community of Kuujjuaq were selected (Figure 4) considering their lithology, quality and extension of the rock exposure, and proximity to the community.

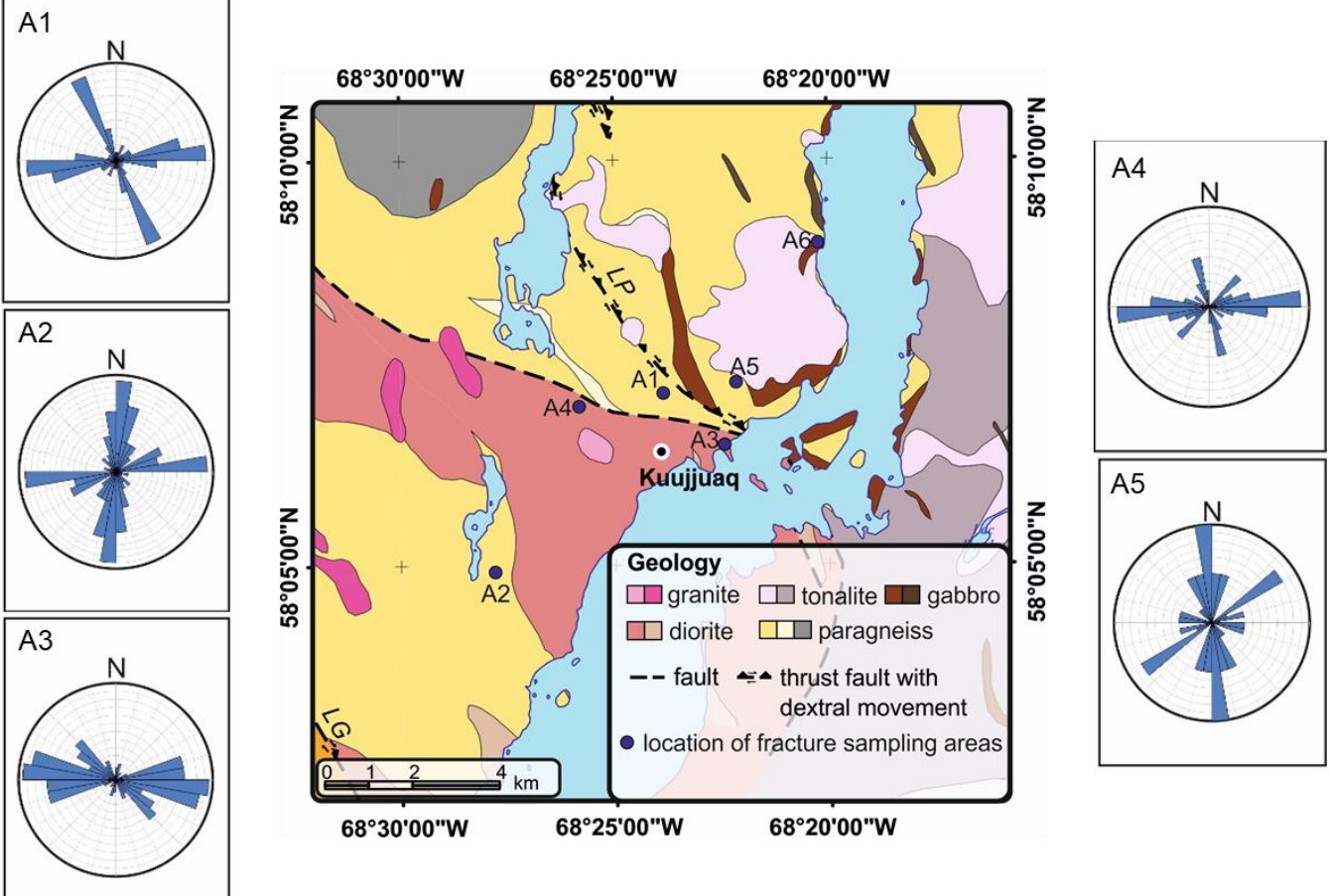

**Figure 4.** Geological map of Kuujjuaq (adapted from [5]) with indication of each fracture sampling area (A1–A5) and rose diagrams produced from data collection. LP—Lac Pingiajjulik fault. The rose diagrams were produced using the software Grapher$^{TM}$ version 19.3.323 26 March 2022, from Golden Software [54].

Sampling areas A1 and A2 are paragneiss outcrops, sampling areas A3 and A4 are outcrops of diorite, sampling area A5 is an outcrop in the paragneiss–tonalite contact, and sampling area A6 is an outcrop showing the contact of tonalite with gabbro.

Sampling area A1 is characterized by two main fracture sets: E-W and NW-SE. Sampling area A2 is also characterized by two main sets: E-W and NNE-SSW. Sampling area A3 is characterized by one main fracture set oriented WNW-ESE and a minor set oriented NW-SE. Sampling area A4 is characterized by one main fracture set oriented E-W and two minor sets oriented NNW-SSE and NE-SW. Sampling area A5 is characterized by two main sets, one oriented N-S and the other oriented NE-SW. In sampling area A6, three main sets were identified: NW-SE, E-W, and NE-SW. Grouping fracture information from these six sampling areas suggests four main fracture sets: F1—E-W; F2—N-S; F3—NNW-SSE; and F4—NW-SE (Figure 5).

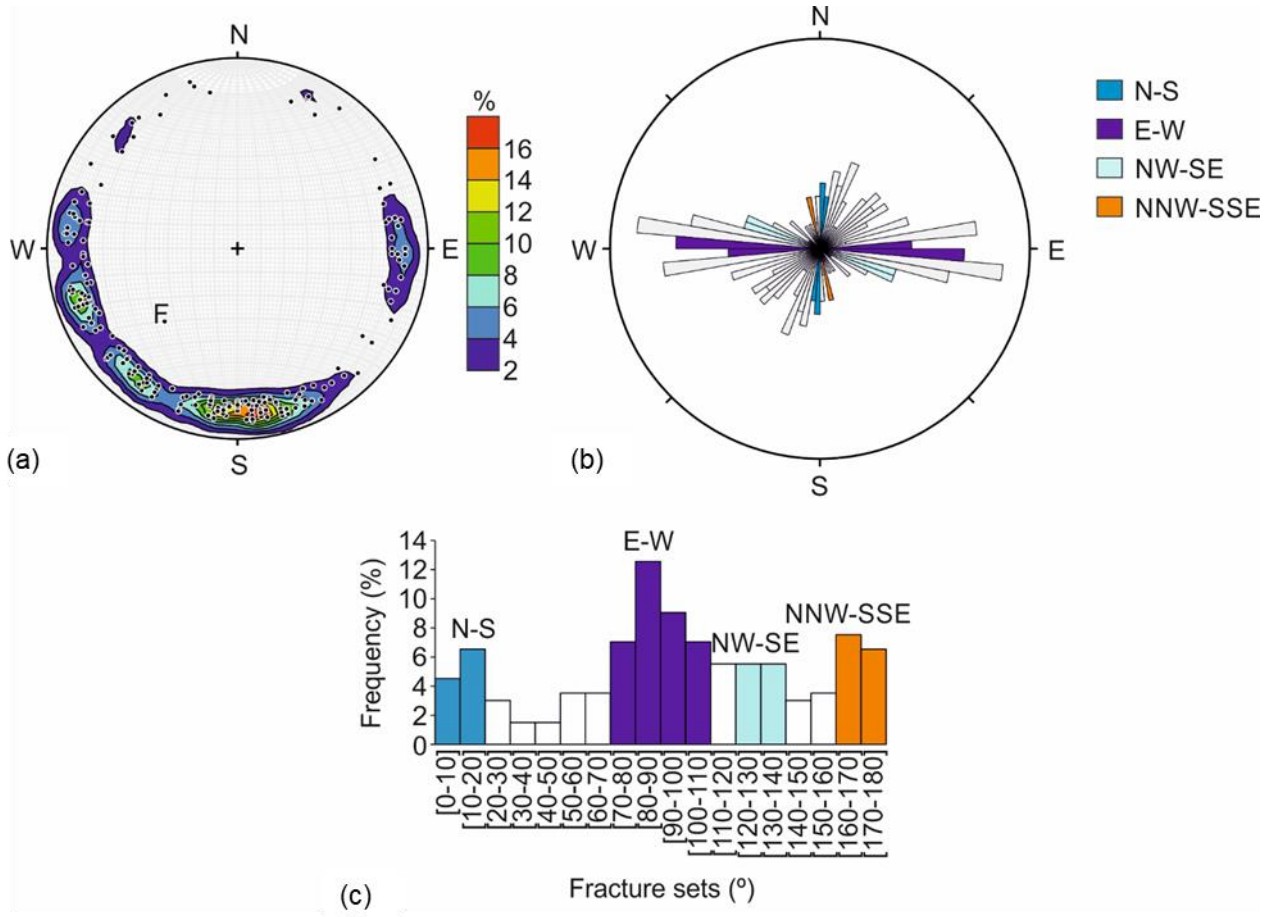

**Figure 5.** (**a**) Stereographic projection of the poles to planes and 1% area contour, (**b**) rose diagram and (**c**) relative frequency of the fracture strike planes. F—Lac Pingiajjulik fault plane. Stereographic projection of the poles to planes and 1% area contour were produced using the software Stereonet version 11.3.7 of Allmendinger et al. [55]. Rose diagram was produced using the software Grapher^TM version 19.3.323 26 March 2022, from Golden Software [54].

A major issue in Kuujjuaq is that the majority of analyzed outcrops have only horizontal surfaces. An exception is outcrop A6 (Figure 6). Here, a vertical surface was available for the analysis, and the dip and dip direction of each fracture could be accurately characterized. The fractures observed suggest a high-angle inclination, with the dip ranging between 60 and 80°.

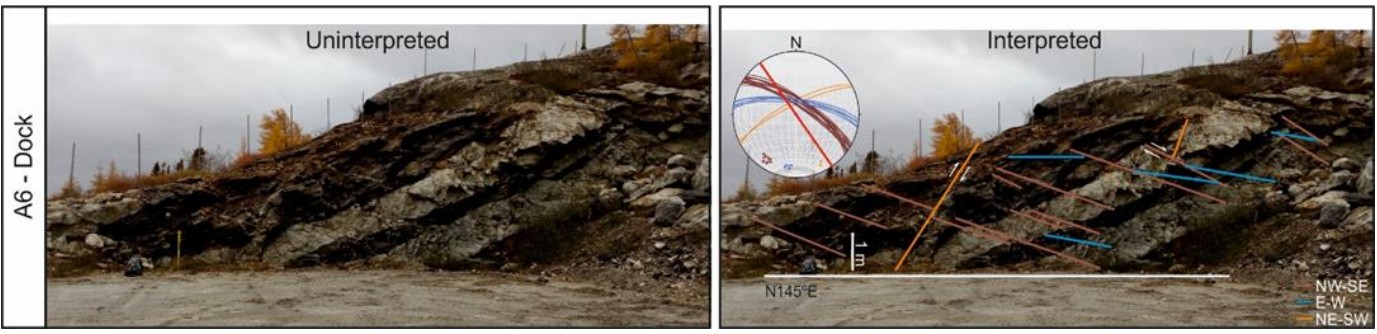

**Figure 6.** Outcrop on sampling area A6 fractures identified and stereonet canvas produced from fracture interpretation [49]. Reprinted/adapted with permission from Ref. [49]. Copyright 2023, The Authors. Red line in the stereonet indicates the outcrop orientation. Stereonet canvas produced using the software Stereonet version 11.3.7 of Allmendinger et al. [55].

### 3. Empirical Stress Regime Estimates

Stress data are almost nonexistent in remote northern regions in Canada (Figure 7a). This, associated with the lack of recorded seismic events (Figure 7b), makes empirical stress predictions a useful tool for a first-order assessment of the activation potential of fractures.

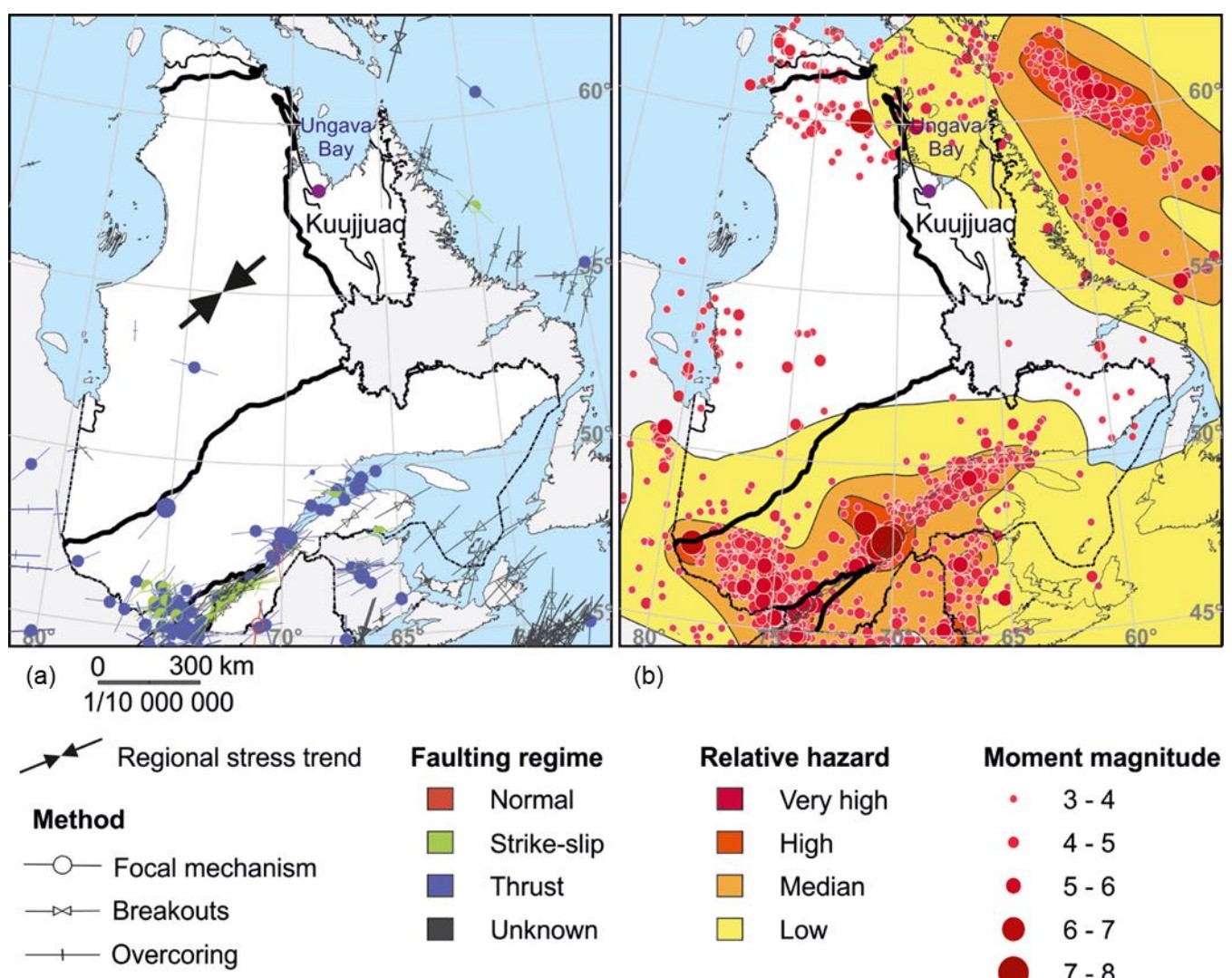

**Figure 7.** (**a**) Stress map [56] and (**b**) seismic hazard map [57] of Québec province [48]. Reprinted/adapted with permission from Ref. [48]. Copyright 2023, The Authors. The arrows in (**a**) indicate the regional trend of the contemporary stress field and the orientation of the maximum horizontal compression based on Adams [58].

The first step in assessing the stress regime through an empirical approach is to compile the available information on the orientation and magnitude of the principal stress components (Table 1). This compilation suggests a regional compression trend of NE–SW for the maximum horizontal stress in the Canadian Shield [58–64], except at the Raglan mining site where the maximum horizontal stress is oriented N–S [64]. The available stress data suggest a reverse faulting regime [58,60,63–65]. The gradient of the vertical stress is found to be about 26.0 MPa km$^{-1}$ [59–66]. The gradient of the minimum horizontal stress ranges between 23.0 and 37.8 MPa km$^{-1}$, while the gradient for the maximum horizontal stress is between 34.4 and 51.3 MPa km$^{-1}$ [59,60,63–66].

**Table 1.** Principal stresses in the Canadian Shield based on the literature review [59–66].

| Principal Stress | Orientation | Magnitude | | Observations | | Ref. |
|---|---|---|---|---|---|---|
| $S_V$ | --- | $(0.0260–0.0324)z$ | $0 < z < 2200$ m | | | |
| $S_{H,average}$ | --- | $9.9 + 0.0371z$ | $0 < z < 900$ m | $S_V < S_{H,average}$ | | [61,62] |
| | | $33.4 + 0.0111z$ | $900 < z\ 2200$ m | | | |
| $S_V$ | --- | $(0.0266 \pm 0.008)z$ | | | | |
| $S_{H,average}$ | --- | $5.9 + 0.0349z$ | $60 < z < 1890$ m | $S_V < S_{hmin} < S_{Hmax}$ | | [59] |
| $S_{Hmax}$ | --- | $8.2 + 0.0422z$ | | | | |
| $S_{hmin}$ | --- | $3.6 + 0.0276z$ | | | | |
| $S_V$ | --- | $0.0285z$ | | | | |
| $S_1$ | N248°/10° | $12.1 + (0.0403 \pm 0.0020)z$ | $0 < z < 2200$ m | $S_V < S_{hmin} < S_{Hmax}$ | | [63] |
| $S_2$ | N300–340°/0° | $6.4 + (0.0293 \pm 0.0019)z$ | | | | |
| $S_3$ | Vertical | $1.4 + (0.0225 \pm 0.0015)z$ | | | | |
| $S_V$ | --- | $0.0260z$ | | | | |
| $S_1$ | NE/horizontal | $13.5 + 0.0344z$ | $0 < z < 6000$ m | $S_V < S_{hmin} < S_{Hmax}$ | | [60] |
| $S_2$ | NW/sub-horizontal | $8.0 + 0.0233z$ | | | | |
| $S_3$ | Vertical | $3.0 + 0.0180z$ | | | | |
| $S_1$ | N-S/horizontal | $0.0513z$ | | | | |
| $S_2$ | E-W/horizontal | $0.0378z$ | --- | $S_V < S_{hmin} < S_{Hmax}$ | | [65] |
| $S_3$ | Vertical | $0.0270z$ | | | | |
| $S_V$ | --- | $(0.0258–0.0263)z$ | | | | |
| $S_1$ | N227°/02° | $(0.040 \pm 0.001)z - (9.2 \pm 1.5)$ | $12 < z < 2552$ m | $S_V < S_{hmin} < S_{Hmax}$ | | [64] |
| $S_2$ | N310°/08° | $(0.029 \pm 0.001)z + (4.6 \pm 1.159)$ | | | | |
| $S_3$ | N270°/88° | $(0.021 \pm 0.001)z - (0.8 \pm 0.872)$ | | | | |
| $S_V$ | --- | $0.021z$ | $0 < z < 1300$ m | | | |
| $S_1$ | --- | $0.012z + 42.4$ | | | | |
| $S_2$ | --- | $0.013z + 24.1$ | $660 < z < 1300$ m | $S_V < S_{hmin} < S_{Hmax}$ | | [66] |
| $S_3$ | --- | $0.007z + 9.7$ | | | | |

$S_V$—vertical stress; $S_{Hmax}$—maximum horizontal stress; $S_{hmin}$—minimum horizontal stress; $S_{H,average}$—average horizontal stress ($\frac{S_{Hmax}+S_{hmin}}{2}$); $S_1$—maximum principal stress; $S_2$—intermediate principal stress; $S_3$—minimum principal stress.

The second step of the empirical approach is applying empirical correlations for a first-order approximation of the magnitude of the principal stresses and in situ fluid pressure.

The vertical stress component can be simply estimated based on the weight of the overlying rock at depth as [67]

$$S_V = \rho g z \tag{1}$$

where $S_V$ (Pa) is the vertical stress, $\rho$ (kg m$^{-3}$) is the density of the geological materials, $g$ (m s$^{-2}$) is the gravitational acceleration, and $z$ (m) is depth.

The horizontal stress components were estimated based on the horizontal to vertical stress ratio as [68]

$$\begin{cases} S_{Hmax} = k_{max}S_V \\ S_{hmin} = k_{min}S_V \end{cases} \tag{2}$$

where $k$ (–) is the stress ratio coefficient and $S_{\text{Hmax}}$ and $S_{\text{hmin}}$ (Pa) are the maximum and minimum horizontal stresses, respectively. Stress ratio expressions obtained for the Canadian Shield (Table 2) were used in this work.

**Table 2.** Horizontal to vertical stress ratios inferred for the Canadian Shield [60–62].

| Stress Ratio Coefficient | Expression | Reference |
|:---:|:---:|:---:|
| $k_{\text{max}}$ | $\frac{357}{z} + 1.46$ | [62] |
| $k_{\text{min}}$ | $\frac{167}{z} + 1.10$ | [62] |
| $k_{\text{max}}$ | $\frac{272 \pm 8}{z} + 1.72$ | [63] |
| $k_{\text{min}}$ | $\frac{30 \pm 4}{z} + 0.86$ | [63] |
| $k_{\text{max}}$ | $7.44z^{-0.198}$ | [60] |
| $k_{\text{min}}$ | $2.81z^{-0.120}$ | [60] |

$k_{\text{max}}$—maximum horizontal to vertical stress ratio; $k_{\text{min}}$—minimum horizontal to vertical stress ratio; $z$—depth in meters.

The in situ fluid pressure can be estimated based on the pore-fluid factor [69,70]:

$$P_{\text{p}} = 0.4 \times S_{\text{V}} \tag{3}$$

where $P_{\text{p}}$ (Pa) is the in situ fluid pressure and 0.4 is the pore to fluid factor assuming a hydrostatic regime.

The Monte Carlo-based sensitivity analysis carried out by Miranda et al. [48] to estimate the magnitude of the principal stresses suggests the following gradients: $27 \pm 1.3$ MPa km$^{-1}$ for the vertical stress, $42 \pm 5.7$ MPa km$^{-1}$ for the maximum horizontal stress, and $30 \pm 3$ MPa km$^{-1}$ for the minimum horizontal stress (Table 3). The gradient for the in situ fluid pressure was estimated as $11 \pm 0.5$ MPa km$^{-1}$ (Table 3). The value on the left side of the $\pm$ sign corresponds to the mean, while the value on the right side of the $\pm$ sign corresponds to one standard deviation. The choice of using gradient values, i.e., MPa km$^{-1}$, instead of absolute values in the analysis was to make the results independent of depth.

Since each stress was estimated individually, the distributions overlap (Figure 8), and some scenarios may be unrealistic (e.g., a normal fault regime). The compilation of stress data measured in the Canadian Shield suggests a reverse faulting regime, with $S_{\text{Hmax}} = S_1$ and $S_{\text{V}} = S_3$ (Table 1). This allows us to eliminate unrealistic scenarios: for example, assuming that the $S_{\text{V}}$ distribution is realistic, then normal faulting regimes can be eliminated. Similarly, scenarios where $S_{\text{hmin}} > S_{\text{Hmax}}$ arises because of the random sampling are rejected.

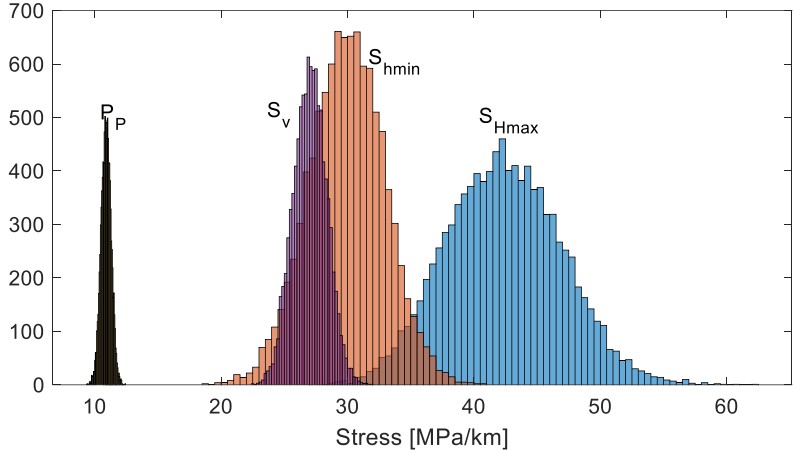

**Figure 8.** Distribution of each principal stress and in situ fluid pressure. $S_{\text{V}}$—vertical stress; $S_{\text{hmin}}$—minimum horizontal stress; $S_{\text{Hmax}}$—maximum horizontal stress; $P_{\text{p}}$—in situ fluid pressure.

**Table 3.** Estimated stress components [47].

| Depth (km) | Statistical Parameters | Principal Stresses | | | |
|---|---|---|---|---|---|
| | | $S_V$ (MPa) | $S_{hmin}$ (MPa) | $S_{Hmax}$ (MPa) | $P_p$ (MPa) |
| 1 | Mean | 27 | 30 | 42 | 11 |
| | St dev | 1.3 | 3.0 | 5.7 | 0.5 |
| | Median | 26.8 | 30.3 | 42.9 | 10.9 |
| | [Min–Max] | [24–30] | [22–38] | [28–56] | [10–12] |
| 2 | Mean | 54 | 57 | 75 | 22 |
| | St dev | 2.6 | 5.0 | 10.2 | 0.9 |
| | Median | 53.7 | 57.0 | 75.7 | 21.7 |
| | [Min–Max] | [48–61] | [44–71] | [49–98] | [19–24] |
| 3 | Mean | 81 | 83 | 107 | 33 |
| | St dev | 3.9 | 6.8 | 14.4 | 1.3 |
| | Median | 80.5 | 83.6 | 108.1 | 32.6 |
| | [Min–Max] | [73–91] | [65–103] | [71–139] | [29–36] |
| 4 | Mean | 108 | 110 | 138 | 44 |
| | St dev | 5.2 | 8.7 | 18.7 | 1.8 |
| | Median | 107.4 | 109.9 | 140.0 | 43.5 |
| | [Min–Max] | [97–122] | [85–136] | [93–185] | [39–48] |
| 5 | Mean | 135 | 136 | 170 | 54 |
| | St dev | 6.6 | 10.7 | 22.7 | 2.2 |
| | Median | 134.2 | 137.1 | 171.2 | 54.3 |
| | [Min–Max] | [121–152] | [107–168] | [113–226] | [49–61] |

$S_V$—vertical stress; $S_{Hmax}$—maximum horizontal stress; $S_{hmin}$—minimum horizontal stress; $P_p$—in situ fluid pressure.

## 4. Slip Tendency Analysis and Reactivation Potential

A well-established method for assessing the fracture slip tendency under different stress conditions is using Mohr diagrams and the Mohr–Coulomb yield criterion [29,71]. The method involves calculating the shear strength of a fracture and comparing it with the maximum shear stress that the fracture can withstand before it slips [71]. The basic equation for the Mohr–Coulomb yield criterion is

$$\tau = c + (\sigma_n - P_p) \times \tan(\phi) \tag{4}$$

where $\tau$ (Pa) is the shear stress acting on the fracture, $c$ (Pa) is the cohesion of the fracture, $\sigma_n$ (Pa) is the normal stress acting on the fracture, $P_p$ (Pa) is the pore pressure in the fracture and $\phi$ (°) is the angle of internal friction of the fracture. $\phi$ is equal to the friction coefficient ($\mu$) as

$$\mu = \tan(\phi) \tag{5}$$

For a cohesionless fracture, i.e., $c = 0$, the equation for the Mohr–Coulomb yield criterion becomes

$$\tau = \mu(\sigma_n - P_p) \tag{6}$$

The fracture is unstable and will slip when the resolved shear stress becomes greater than the effective normal stress multiplied by the friction coefficient acting on the fracture plane:

$$\tau > \mu(\sigma_n - P_p) \tag{7}$$

Meeting this condition generally requires the fracture to be hydraulically connected to a source of pore pressure perturbation but also mechanically unstable, potentially leading to the generation of seismic events as the slip takes place. The shear and normal stress acting on a fracture plane can be estimated using mathematical equations that take into account the stress tensor and the orientation of the fracture [71].

Figure 9 shows the 3D Mohr diagram produced considering the average stress magnitude gradients based on the empirical stress regime estimates described in the previous section ($S_{\text{Hmax}}$ 42.1 MPa km$^{-1}$, $S_{\text{hmin}}$ 30 MPa km$^{-1}$, $S_{\text{v}}$ 27 MPa km$^{-1}$).

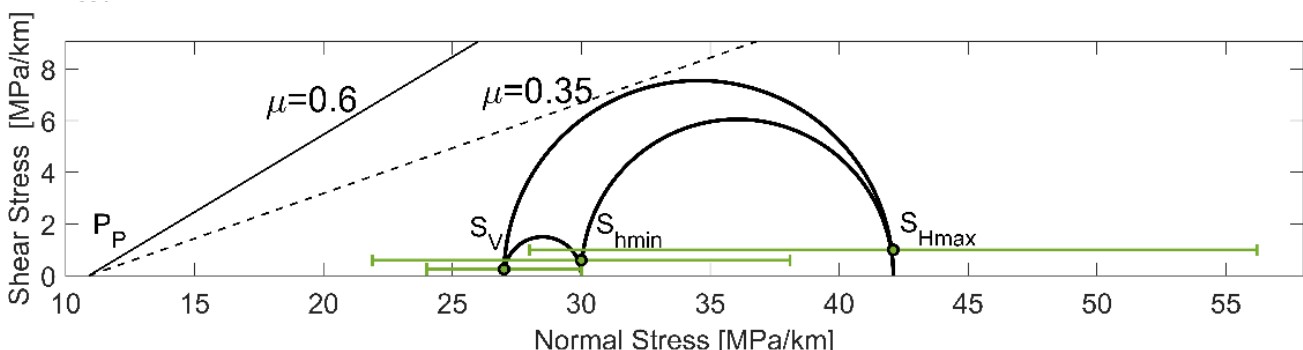

**Figure 9.** A 3D Mohr diagram for the average stress magnitude gradient and uncertainty associated with each principal stress component.

A point in the 3D Mohr diagram can represent a fracture plane with shear stress ($\tau$) and normal stress ($\sigma_{\text{n}}$) acting on it. The stability of the plane can be assessed by comparing the shear and normal stresses to the Mohr–Coulomb yield criterion. For the case in our study, with respect to the average stress magnitude gradient, the minimum required coefficient of friction is 0.35 (Figure 9). This indicates that the ratio of shear stress ($\tau$) to normal stress ($\sigma_{\text{n}}$) acting on the most likely fracture to slip should be less than 0.35 for the rock mass to remain stable.

This coefficient of friction is smaller than the typical coefficient of friction for joint and fault surfaces in crystalline rock; usually it ranges between 0.6 and 1.0 [72]. Considering the minimum required coefficient of friction is 0.35, most fractures are considered stable, meaning they are not prone to slip under the existing stress conditions. However, the stability of fractures is contingent upon the state of stress remaining constant. If the stress conditions change, for example through thermoelastic processes as well as pore pressure changes, the stability of fractures can be altered [73].

The pore pressure parameter in the Mohr–Coulomb yield criterion can be changed by human intervention and thus alter the state of stress surrounding the fractures. This occurs during hydraulic stimulation techniques used to increase the productivity of geothermal reservoirs. Fluid injection increases the pore pressure in the surrounding interconnected fractures, resulting in a reduction of the effective normal stress, thereby allowing stressed fractures in the right orientation to slip more easily (Equation (9)). Furthermore, hydraulic stimulation of the rock mass is intended to create new pathways for fluid flow [22], so that a pore pressure perturbation can affect more distant fractures and potentially a critically stressed fault. As the flow efficiency of the geothermal reservoir is improved by creating better fracture conductivity in the natural fracture system, more structures are involved in the pressure perturbation.

The critical pore pressure perturbation to trigger slip is expressed as

$$\Delta P = P_{\text{injection}} - P_{\text{p}} \tag{8}$$

where $P_{\text{injection}}$ (Pa) is the fluid pressure of the injected fluid.

Figure 10, displayed as a lower hemisphere stereonet, provides a visual representation of the changes in pore pressure required for fractures to initiate slip considering the most likely stress regime. Each point on the stereonet corresponds to a fracture in a particular orientation; the color indicates the magnitude of the necessary pore pressure perturbation for slip initiation. Fractures in the red regions require the smallest $\Delta P$ perturbation to cause slip, while those in blue tones require the largest perturbation.

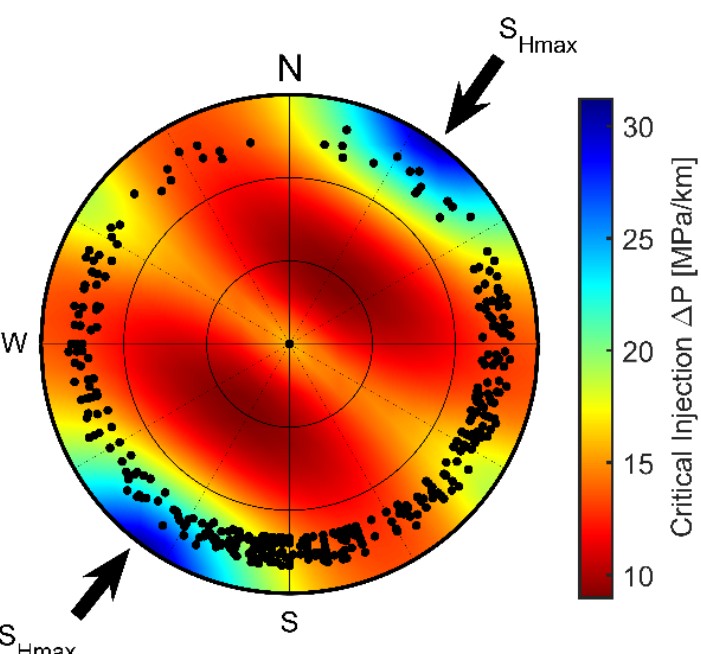

**Figure 10.** Lower hemisphere stereonet showing critical injection pressure perturbation leading to fracture slip. Each point in the graph represents a fracture pole identified in the field.

The stereonet suggests that fractures oriented perpendicular to the maximum horizontal stress with a low dip angle, i.e., $30 \pm 10°$, are under the most critical stress conditions. Specifically, for this scenario, the injection pressure should be approximately $\Delta P = 10$ MPa km$^{-1}$ higher than the current estimated in situ fluid pressure of 10.9 MPa km$^{-1}$.

However, the fractures observed in the field have high dip angles (60–80°), making them less critically stressed. Optimally oriented sets that require less injection fluid pressure to be activated are WNW-ESE and N-S ($\Delta P$ about 12.5 MPa km$^{-1}$). The fracture sets striking E-W and NE-SW would require a $\Delta P$ of about 17.5 MPa km$^{-1}$ to induce slip. The least slip-sensitive set is NW-SE, which is oriented parallel to the minimum horizontal stress.

The slip tendency of a fracture is expressed as

$$\frac{\tau}{(\sigma_n - P_p)} \geq \mu \tag{9}$$

In a Mohr–Coulomb shear slip assessment, different input parameters (e.g., stress, friction angle) can be treated as random variables with specific statistical parameters. The probabilities of slip can then be described as

$$P_{\text{failure}} = P(\tau - (\sigma_n - P_p) \times \mu \leq 0) \tag{10}$$

This probabilistic slip tendency approach becomes particularly useful when dealing with large uncertainty in each input parameter. As previously mentioned, and discussed by Miranda et al. [48], there are uncertainties associated with the magnitude and orientation of each principal stress as well as the value of pore pressure. These uncertainties associated with the magnitude of three principal stresses are shown by the green error bars in Figure 9. Uncertainties in the dip direction and dip angle of fractures also exist. A joint/fault is seldom a straight discontinuity plane: it can have different shapes and orientations, ranging from straight or planar fractures to irregular or curved cracks. The latter have even lower slip tendencies because of their macroscopic roughness (deviation from planarity).

To account for these uncertainties in stress, pore pressure and fracture properties, we utilized Monte Carlo probabilistic methods to generate a range of possible outcomes. The Mohr–Coulomb yield criterion was then applied to each of these outcomes to determine

the probability of fracture slip. This approach allowed for a more comprehensive analysis of the potential for fracture slip, considering the different uncertainties in the system. Figure 9 and Table 3 present distributions of three principal stress magnitudes and pore pressure as inputs to the analysis. Recall that a constraint was imposed to neglect unrealistic stress scenarios. Our analysis assumes that the friction coefficient ranges between 0.5 and 0.7 and that fracture dip direction and dip angle deviate by up to 5° compared to the field measurement. This analysis was conducted using 10,000 random combinations of parameters for each mapped fracture segment to assess the conditional probability of slip as a function of pore pressure perturbation ($\Delta P$).

Using the uncertainty distributions and fracture properties as randomly selected inputs, the distribution of critical injection pressure calculated from 10,000 scenarios shows a wide range (Figure 11). The curve shows the cumulative probability of exceeding the critical injection pressure, with higher probabilities indicating a greater likelihood of fracture slip. The distribution can be used to estimate the range of injection pressures that are likely to cause fracture slip, as well as the probability of such events occurring under different injection scenarios. Results of the analysis indicate a very low probability of fracture slip that increases with increasing pore pressure. However, when the injection pressure is raised to 26 MPa/km, the probability of slip increases to 68%, which further escalates with a higher injection pressure.

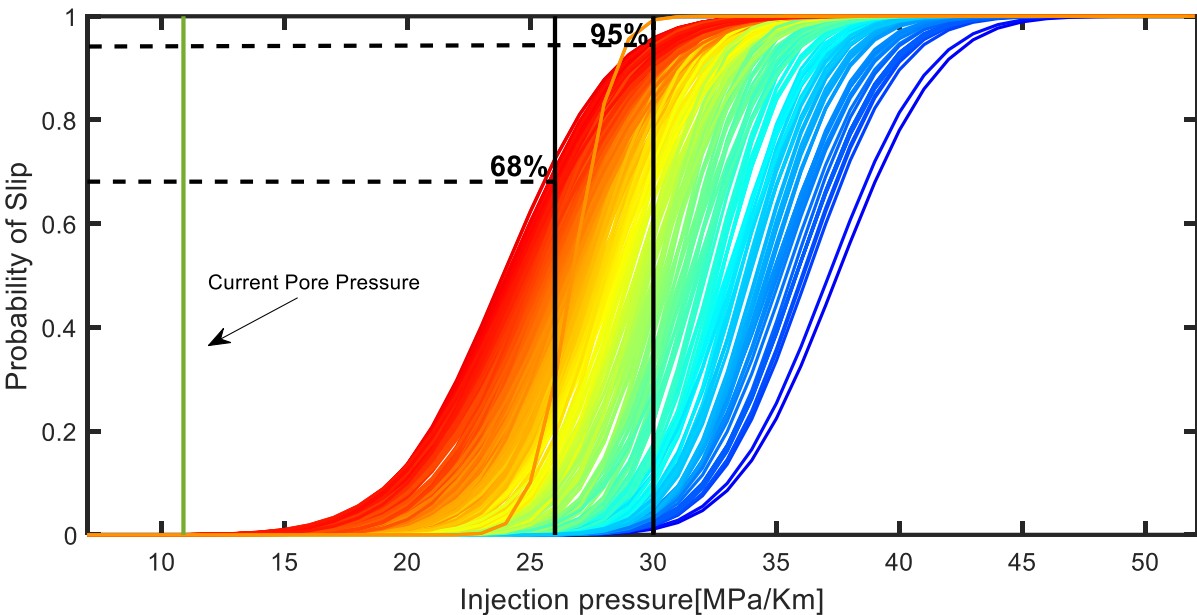

**Figure 11.** Each curve represents the cumulative probability function of slip on each fracture at current pressure. The hot color curve represents fractures with a higher slip tendency. The higher the injection pressure, the more likely the fractures are to slip. An injection pressure of 30 MPa/km is expected to result in a 95% fracture slip.

Tornado plots were used to visualize the sensitivity of injection pressure to slip for each input parameter to evaluate the impact of uncertainties in specific parameters on the slip tendency of a particular fracture plane. The parameters considered are ranked based on the extent of variation in the resulting pore pressure to induce slip. The magnitude of the variation indicates which parameters exert the most significant influence on the calculated pore pressure required to induce slip. Figure 12 is an example of a tornado plot, focusing on the fracture that has the highest slip tendency. For this specific case, the fracture is particularly sensitive to variations in the $S_{hmin}$ gradient, as indicated by the error bar associated with $S_{hmin}$ in Figure 9. It is important to note that the sensitivity analysis may vary for each fracture, depending on its unique strike and dip angle.

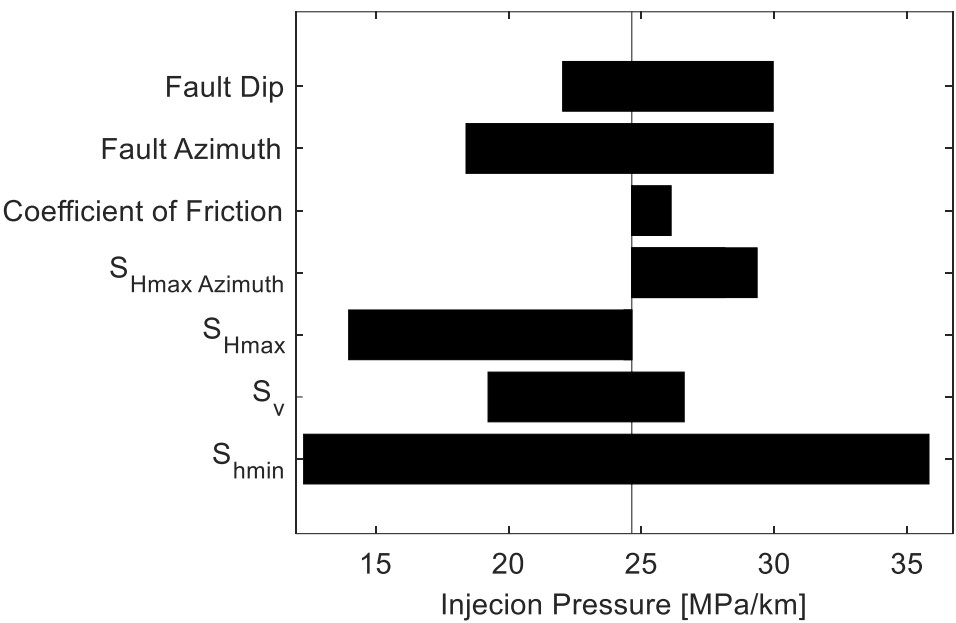

**Figure 12.** Tornado plot displaying a sensitivity analysis of uncertain parameters affecting the slip tendency of the optimally oriented fracture.

## 5. Discussion

The EGS is an emerging technology that could revolutionize energy supply in remote northern communities located on Canadian Shield rocks. However, the EGS is still a challenging technology with many development issues that need to be addressed. Pollack et al. [74] compiled challenges experienced in world-wide EGS projects and found that:

- 6 sites ceased operations temporarily or shut down due to seismicity or seismicity concerns;
- 24 EGS projects were delayed or terminated due to drilling and plant operation issues, such as holes and cracks in wellbore casing, stuck drill strings, and well collapses;
- 18 sites faced challenges in reservoir creation and circulation, such as insufficient connectivity between the injection and production wells or water loss.

Despite all these potential roadblocks, Pollack et al. [74] also found that 29 sites had successful stimulation experiences, and these sites are active and improving today's knowledge about EGS technology.

The main objective of this technology is to develop a low-impedance hydraulic connection between injector and producer wells that will allow reasonable flow rates, minimal thermal drawdown, minimal water losses, and induced seismicity at levels imperceptible to the population. For these reasons, it is important to forecast how fractures will behave during hydraulic stimulation treatments. This is tightly linked to the natural fracture network properties at the target site, the in situ stress field and the friction properties of the fractures.

Slip tendency analysis is a simple yet effective tool to answer questions that arise at early stages of geothermal exploration:

1. What is the probability of shear slip of pre-existing fractures at the current state of stress?
2. Which orientations of fractures are most likely to be activated?
3. What in situ fluid pressure is required to overcome the shear stress and activate pre-existing fractures?

Considering the results of our analysis, field fracture observations, and empirical stress regime estimates, none of the fractures at the target site are near a critical state of stress, meaning a very low (<1%) probability of slippage under the current state of stress. Nevertheless, if pore pressures change (i.e., hydraulic stimulation treatments or higher

circulation pressures), the fracture sets making a 30° angle with the orientation of the maximum horizontal stress (i.e., WNW-ESE and N-S) may be reactivated at an injection pressure ($\Delta$P) of about 12.5 MPa km$^{-1}$. A $\Delta$P of about 17.5 MPa km$^{-1}$ may be necessary to reactivate the fracture sets parallel to the maximum horizontal stress (i.e., NE-SW) and the E-W fracture set. The least optimally oriented set is the NW-SE fracture set, which is oriented parallel to the minimum horizontal stress (depicted in blue in Figure 10). Again, it is important to emphasize that these estimates are empirical, and several factors influence how fractures behave during hydraulic stimulation. A more robust analysis including field experiments would be necessary to validate our estimates.

This observation is somewhat in line with what was found with the first hydraulic stimulation experiments at the Carnmenellis Granite, Cornwall, UK [25]. In this EGS site, micro-seismicity patterns indicated anisotropic fracture-controlled flow closely related to the orientation of the maximum horizontal stress [25]. It was deduced that, due to the impact of the anisotropic stress field on fracture apertures, the preferential flow paths were fractures favorably oriented to the maximum horizontal stress. These tend to be activated in shear or in transtension. At Kuujjuaq, our study site, a detailed structural analysis would be the next logical step to verify if a relationship between the fracture aperture and orientation exists.

Additionally, it is important to emphasize that the injection pressure values to activate the fractures are only valid if the most likely scenario in terms of stress prevails in the study area. In instances where the differential stresses are smaller or larger than the average value, the required activation injection pressure is different.

In a high differential stress scenario ($\sigma_1 - \sigma_3$ = 29 MPa km$^{-1}$), the optimally oriented fracture is near their critical stress state, and an injection pressure of $\Delta$P ~ 2 MPa km$^{-1}$ would likely lead to slip, and induced seismicity would be far more likely. However, in a low differential stress scenario ($\sigma_1 - \sigma_3$ = 9 MPa km$^{-1}$), this fracture would be activated at injection pressures of $\Delta$P ~ 24 MPa km$^{-1}$ (Figure 11).

The influence of differential stress on hydraulic stimulation effectiveness has been highlighted by Xie et al. [22] who compiled data from seven EGS sites and noticed that where the differential stresses were large, less additional fluid pressure ($\Delta$P) was required to activate shear slip of natural fractures. Their observations showed that the Hijiori EGS project (Japan) is the least stressed and has the smallest differential stress. For this reason, it required the most effort to trigger shear slip compared to, for instance, Rosemanowes (UK), Soultz (France), and Basel (Switzerland) EGS projects. In other words, when the differential stress is small, more fluid pressure is needed to move the Mohr stress circle to meet the slip yield envelope.

It is also important to mention the role played by the intermediate principal stress. The activation of pre-existing fracture planes is dependent on not only the change in the maximum differential stress, but also the value of the intermediate stress. For example, if the intermediate stress was equal to the minimum stress instead of equal to the maximum stress in the worst-case stress scenario, then the required injection pressure to reactivate the fractures would be less than 24 MPa km$^{-1}$. Similarly, if the intermediate stress was larger than assumed in the best-case stress scenario, then the fractures would not be at a critical state of stress and an injection pressure of about 11 MPa km$^{-1}$ would be necessary to reactivate them.

The influence of external factors, such as an injection of cold water into a warm medium, can also alter the state of stress, increasing the deviatoric stress. If this increases the shear stress along favorably oriented discontinuities while reducing the normal stress, the fractures move toward a more critical condition. This study only takes into account pressure changes, but further research can be undertaken to consider temperature changes as well.

Another aspect worth discussing is the lithological heterogeneity. The study area is composed mainly by paragneiss rocks. However, diorite, gabbro, tonalite, and granites can be found interlayering the paragneiss. Some rock types may exhibit higher or lower

slip activation potential, and this could cause a heterogeneity in the system that would change the analysis undertaken. To avoid adding more uncertainty sources to the analysis, we considered the system to be homogeneous in terms of lithology. However, further research is needed and should be undertaken to deepen the simple analysis carried out in this work. Additionally, anisotropy was also not considered, but it could be included in a more complex model.

The behavior of the Lac Pingiajjulik fault (see Figure 3 for location) during hydraulic stimulation is another question that deserves further investigation. This corresponds to a large-scale shear zone, and one of two scenarios can happen when it is stimulated: either it acts as a preferential flow pathway, or it acts as a barrier to flow circulation. In either case, the EGS must be properly designed to meet the goal of high flow rates, minimal thermal drawdown, minimal water losses, and induced seismicity at levels imperceptible to the population.

Hydraulic shearing experiments were conducted at the Grimsel Test Site in Switzerland under the framework of the In-situ Stimulation and Circulation experiment [15,16]. The rock volume tested is intercepted by two sets of shear zones of ductile (S1.0–S1.3) and brittle-ductile (S3.1–S3.2) origin [75]. The first set of shear zones (S1.0 to S1.3) includes three ductile shear zones, which show strong foliation and mylonitization. The second set (S3.1 and S3.2) includes two shear zones that bind a densely fractured zone in between.

The hydraulic stimulation experiments reveal that fractures associated with one out of two shear zone types were hydraulically reactivated [76]. The authors also noted that the two shear zone types not only differ in terms of tectonic genesis and architecture, but transmissivity change, jacking pressure, and seismic activity were also different for the two shear zone types. These observations led them to suggest that shear zone architectures govern the seismo-hydromechanical response. Based on the in situ experiments, Krietsch et al. [76] observed that elevated fracture fluid pressures associated with the stimulations propagated mostly along the stimulated shear zones, and that flow is channelized within the shear zones. Furthermore, Villiger et al. [77] observed three seismic clusters associated with the stimulation of fractures in the damage zone of the shear zone.

The hydraulic stimulation experiments carried out at the Grimsel Test Site highlight the need to properly characterize the tectonic genesis and architecture of the shear zone crossing Kuujjuaq. Further structural analysis of the area is the next logical step to gain insights into its behavior.

It is also worth discussing how useful hydraulic experiments could be to decrease the uncertainty found in this research work. Hydraulic stimulation experiments and microseismic monitoring are nowadays relatively feasible and could provide important insights into the reservoir development. However, carrying out these field experiments is quite challenging in northern and remote environments where there is no road linking the southern and northern areas and the equipment needs to be shipped by plain or boat. A first-order assessment in this situation is necessary to trigger interest for further investments in geothermal research.

## 6. Conclusions

Crystalline rocks can host important geothermal resources, but the permeability of these reservoirs is usually too low for the system to be commercially viable. The EGS is an emerging technology that can help increase the productivity of reservoirs by enhancing hydraulic connections within the fracture network. However, the EGS is still a challenging technology: issues associated with reservoir development and circulation and induced seismicity are the main roadblocks to the successful deployment of this technology.

Therefore, understanding how the fracture network will behave when hydraulically stimulated is a key step toward the development of the EGS. The objective of this work was to estimate the slip activation potential of existing fractures in subsurface crystalline basement rocks from outcrop analogues in the Kuujjuaq area and empirical stress predictions.

With this work, we addressed questions that arise at early stages of geothermal exploration, namely "What is the probability of shear slip of the pre-existing fractures at the current state of stress?", "Which orientations of fractures are most likely to be activated?", and "What in situ fluid pressure is required to overcome the shear stress and activate pre-existing fractures?".

Our analysis using a probabilistic slip tendency approach suggests that, at the current state of stress, the probability of the pre-existing fractures slipping is very low (<1%). The analysis also suggests that the optimally oriented fracture sets that can be activated at lower injection pressures are WNW-ESE and N-S. An injection pressure of 12.5 MPa km$^{-1}$ could be necessary to activate shear slip; however, this is only valid considering the most likely stress scenario. If the differential stress is larger or smaller than the average scenario, then larger or smaller injection pressures are required to reactivate the optimally oriented fractures.

The next logical and needed steps should include more accurate stress measurements than the empirical stress estimates used in this work and a comprehensive structural characterization of the joints, faults, and shear zones in the study area. Hydraulic stimulation experiments would also be needed to not only gain insights into the reservoir and its behavior when pressurized, but also to properly design the EGS to meet the targets of high flow rates, minimal thermal drawdown, minimal water losses, and induced seismicity at levels imperceptible to the population.

**Author Contributions:** Conceptualization, M.M.M. and A.Y.; methodology, M.M.M. and A.Y.; software, A.Y.; validation, M.M.M., A.Y., J.R., A.W. and M.B.D.; formal analysis, M.M.M. and A.Y.; investigation, M.M.M. and A.Y.; resources, M.M.M. and A.Y.; data curation, M.M.M. and A.Y.; writing—original draft preparation, M.M.M. and A.Y.; writing—review and editing, M.M.M., A.Y., J.R., A.W. and M.B.D.; visualization, M.M.M. and A.Y.; supervision, J.R. and M.B.D.; project administration, J.R.; funding acquisition, J.R. All authors have read and agreed to the published version of the manuscript.

**Funding:** This research was funded by the Institut Nordique du Québec (INQ) through a research chair awarded to Jasmin Raymond to evaluate the geothermal potential of northern Québec.

**Data Availability Statement:** All data are presented in the manuscript.

**Acknowledgments:** The Centre d'études nordiques (CEN), supported by the Fonds de recherche du Québec—nature et technologies (FRQNT), and the Observatoire Homme Milieu Nunavik (OHMI) are acknowledged for helping with field campaigns and logistics. The authors are also thankful to Cynthia Brind'Amour-Coté for her support during the field campaign. The authors would also like to acknowledge the two anonymous reviewers whose comments helped to improve the manuscript.

**Conflicts of Interest:** The authors declare no conflict of interest. The funders had no role in the design of the study; in the collection, analyses, or interpretation of data; in the writing of the manuscript; or in the decision to publish the results.

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
