# Peer review of "Slip Activation Potential of Fractures in the Crystalline Basement Rocks of Kuujjuaq (Nunavik, Canada) to Assess Enhanced Geothermal Systems Development"

_geosciences, doi:10.3390/geosciences13110340_

Round 1

Reviewer 1 Report

Comments and Suggestions for Authors

The study aims to estimate the slip activation potential of pre-existing fractures in the crystalline rocks of Kuujjuaq, Canada, using outcrop analogues and theoretical stress regime estimates. It employs a probabilistic slip tendency approach to discern the most likely fracture orientations to activate under hydraulic stimulation, a key aspect in Enhanced Geothermal Systems (EGS). It concludes that the probability of pre-existing fractures slipping under the expected conditions is less than 1%, suggesting that the rock formations are relatively stable. The study leverages outcrop analogues for its geological assessment, a common but sometimes limiting approach. The theoretical stress regime estimates, rather than empirical measurements, are used to evaluate slip activation potential. This is a significant limitation, as theoretical models often deviate from real-world conditions. The study fills an important gap by applying probabilistic slip tendency analysis to EGS in a region that could benefit from geothermal development. However, the relatively simple approach and the use of theoretical models may not capture the full complexity of geological and geomechanical processes.

Limitations

The use of theoretical stress estimates may not reflect the actual stress conditions.

The study lacks a comprehensive structural characterization of joints, faults, and shear zones, which would be pivotal in a complete assessment.

Major Questions:

  1. Why were theoretical stress models preferred over empirical stress measurements?

  2. How do you account for uncertainties in the orientation of the fracture sets?

  3. Is the 12.5 MPa km−1 injection pressure universally applicable across all optimally oriented fractures?

  4. Are there any specific rock types within the crystalline basement that exhibit higher or lower slip activation potentials?

  5. Does the study consider anisotropy in the mechanical properties of the crystalline rocks?

  6. How were the geological outcrop analogues selected, and do they accurately represent the sub-surface conditions?

  7. What is the sensitivity of the probabilistic model to variations in stress and fluid pressure?

  8. Could the model be validated with hydraulic stimulation experiments to verify the findings?

Minor Questions:

  1. Is the 1% slip activation potential estimated applicable across seasons or is it sensitive to temporal variations?

  2. How does the study account for naturally occurring seismic events in its calculations?

  3. What was the range of dip and strike measurements and their statistical distribution?

  4. Could external factors like temperature fluctuations impact the stability of fractures?

  5. Does the study consider shear versus tensile failure modes in the fractures?

  6. Is the study area homogeneous or does it possess sub-regions with varying geological characteristics?

The study makes a significant contribution by providing a baseline assessment of slip activation potentials, which is vital for the development of EGS in the Kuujjuaq area. However, its reliance on theoretical stress models and absence of comprehensive geological characterization warrant cautious interpretation of the results. Further corrections should aim to incorporate empirical stress measurements and detailed structural characterizations to improve the model's predictive accuracy.

Author Response

Dear reviewer,

Thank you for your thorough reading of the manuscript and comments. The quality of the manuscript was greatly improved. In the next lines, point-by-point explanations are given to address each comment made.

The study aims to estimate the slip activation potential of pre-existing fractures in the crystalline rocks of Kuujjuaq, Canada, using outcrop analogues and theoretical stress regime estimates. It employs a probabilistic slip tendency approach to discern the most likely fracture orientations to activate under hydraulic stimulation, a key aspect in Enhanced Geothermal Systems (EGS). It concludes that the probability of pre-existing fractures slipping under the expected conditions is less than 1%, suggesting that the rock formations are relatively stable. The study leverages outcrop analogues for its geological assessment, a common but sometimes limiting approach. The theoretical stress regime estimates, rather than empirical measurements, are used to evaluate slip activation potential. This is a significant limitation, as theoretical models often deviate from real-world conditions. The study fills an important gap by applying probabilistic slip tendency analysis to EGS in a region that could benefit from geothermal development. However, the relatively simple approach and the use of theoretical models may not capture the full complexity of geological and geomechanical processes.

Limitations

The use of theoretical stress estimates may not reflect the actual stress conditions.

The study lacks a comprehensive structural characterization of joints, faults, and shear zones, which would be pivotal in a complete assessment.

Major Questions:

  1. Why were theoretical stress models preferred over empirical stress measurements?

The stress estimate approach followed in this work is not theoretical, but rather empirical. It leans on previous measurements and empirical relationships for extrapolation. The term theoretical was used in the previous version of the manuscript to explain that the estimates are not from field experiments but rather from literature review and empirical relationships. We understand now that using the term theoretical is misleading, and it was changed to empirical throughout the manuscript.

  1. How do you account for uncertainties in the orientation of the fracture sets?

We considered that fracture dip direction and dip angle deviate by up to 5° compared to the field measurement for the probabilistic slip tendency analysis. However, the model is versatile enough to consider more than 5° uncertainty. This is mentioned in section 4.

  1. Is the 12.5 MPa km−1 injection pressure universally applicable across all optimally oriented fractures?

This is the pressure at the wellhead. The pressure at depth is the sum of the hydrostatic head in the pore fluid at depth and the wellhead pressure. The salinity of the pore fluid increases with depth, reaching a value in the range of 1.2 gm/cc (NaCl saturated brine) at some depth that is not well defined in Nunavik, but has been studied in Ontario in the context of deep nuclear waste repository design. However, the distribution of the density with depth, going from fresh water to saturated brine at some unspecified depth, remains unknown. Also, there may be substantial amounts of CaCl2 and MgCl2 in the deep crystalline waters, giving a higher density. Finally, it is not clear what the system permeability is at depth; it might be so low that the functional natural pore pressure is irrelevant. In that case, the pressure at depth for a pressurization even is the hydrostatic head of the wellbore fluid plus the surface static gauge pressure. There is obviously a lot of uncertainty, and a “robust empirical estimate” (not a theoretical one!) is appropriate.

  1. Are there any specific rock types within the crystalline basement that exhibit higher or lower slip activation potentials?

Of course. If there is a perfectly oriented fault or extensive joint (or cohesionless foliation surface) in a greenschist, phyllite or schist, a lower friction angle might be postulated. However, by assuming c = 0, and by assuming planarity, extremely conservative predictions have already been undertaken. A more detailed first-order assessment in the absence of deep lithological and structural data is not warranted, in our opinion. It would render the analysis more complex, and under conditions of substantial uncertainty, not contribute substantively to the discourse. A paragraph was added to the discussion highlighting this question (lines 580-587).

  1. Does the study consider anisotropy in the mechanical properties of the crystalline rocks?

Anisotropy inclusion is not necessary because we are not estimating change in mechanical stress through a poroelastic geomechanical model.  In any case, the relevance of a poroelastic model in a rock mass that is characterized by tight joints and faults is indeed questionable, as the propagation of increased pressure into the rock blocks (or even through the uncharacterized rock mass) is a matter for gross speculation.  Again, a more complex model is not appropriate. This was added to the discussion (line 587-588).

  1. How were the geological outcrop analogues selected, and do they accurately represent the sub-surface conditions?

Rock is exposed at surface in most of the study area and is believed to represent the subsurface lithologies. We selected areas to sample based on their proximity to the community, quality of the rock exposure, extension of the rock exposure, and representative of the main lithologies. This was added to section 2 (lines 208-209).

  1. What is the sensitivity of the probabilistic model to variations in stress and fluid pressure?

We conduct a sensitivity analysis by systematically varying stress and other inputs within their respective ranges or uncertainty bounds while keeping other model inputs constant. This process is illustrated in Figure 12 using tornado plots (lines 473-485). It's important to note that the results of sensitivity analysis may vary for different fractures due to their distinct dip directions and dip angles.

  1. Could the model be validated with hydraulic stimulation experiments to verify the findings?

The uncertainty in the model can be reduced as more data become available at depth, including stress measurements, deconvolution of the rock mass fabric and the presence of low-cohesion faults at depth, and so on.   Stress estimates can be refined by careful hydraulic fracturing testing, but even this gives results that are often quite constrained in terms of uncertainty with respect of the “other” two stresses. Especially if sigma 3 is vertical, which we are almost certain for some depth from the surface.    

In the course of an actual project, microseismic monitoring during injection experiments is quite feasible, and this may serve as a means of validating the model.  In the absence of actual deep data, speculation is inappropriate. A sentence was added to the discussion explaining this question (lines 618-625).

Minor Questions:

  1. Is the 1% slip activation potential estimated applicable across seasons or is it sensitive to temporal variations?

We have not taken into account the seasonal effect in our analysis, although it is likely minimal due to the independence of geomechanical parameters at depths exceeding 1 km from surface parameters. Additionally, it's worth noting that the impact of thermoelastic and poroelastic effects has not been included in our analysis, as mentioned in our paper.

  1. How does the study account for naturally occurring seismic events in its calculations?

The region is extremely quiet seismically as illustrate by Figure 7. However, spatiotemporal clustering analysis will allow discrimination.

  1. What was the range of dip and strike measurements and their statistical distribution?

Both dip and strike distribution are shown in Figure 5.

  1. Could external factors like temperature fluctuations impact the stability of fractures?

Yes: the long-term injection (circulation) of a cold fluid will alter the stress distributions and increase (in general) the deviatoric stresses in the ground.  If this increases the shear stress along favorably oriented discontinuities while reducing the normal stress, they move toward a more critical condition.  However, this study is focussed on pressure changes, not temperature changes. A paragraph was added to the discussion to include this question (lines 575-579).

  1. Does the study consider shear versus tensile failure modes in the fractures?

If a planar discontinuity is favorably oriented with respect to the stress conditions and cohesion is assumed to be zero, slip before tensile opening of the discontinuities is generally favored, but there are many factors that can affect slip such as slip plane roughness scale.  Tensile parting is at the very least equal to a small bottom-hole pressure increment above sigma 3, but slip can, on favorably oriented discontinuities without cohesion, take place at injection pressures below this (the Mohr-Coulomb frictional model assumption).  Furthermore, tensile parting is inconsequential because it does not lead to substantial seismic energy emission, whereas stick-slip events do, and they can be large if the discontinuity plane (shear slip surface) is all close to criticality and is smooth enough so that slip propagates along the plane.

  1. Is the study area homogeneous or does it possess sub-regions with varying geological characteristics?

All rock masses have inhomogeneity at all scales (crystalline scale to 1000’s of kilometers).  This is a first-order assessment in the absence of deep data, so a reduced yet robust model is best.

The study makes a significant contribution by providing a baseline assessment of slip activation potentials, which is vital for the development of EGS in the Kuujjuaq area. However, its reliance on theoretical stress models and absence of comprehensive geological characterization warrant cautious interpretation of the results. Further corrections should aim to incorporate empirical stress measurements and detailed structural characterizations to improve the model's predictive accuracy.

Reviewer 2 Report

Comments and Suggestions for Authors

 Slip activation potential of fractures in the crystalline basement rocks of Kuujjuaq (Nunavik, Canada)

General comments:

The paper is related to the computation of the probability of slip activation in EGS systems using the Montecarlo simulation method.

As general comment, the title doesn’t reflect the content of the paper. I suggest modifying it by adding “enhance geothermal energy”, for example.

Also, the abstract doesn’t reflect the rest of the paper. The abstract must be carefully rewritten as it is not clear and does not introduce the current work adequately.

The results presentation is weak (just Figure 11, which also needs improvement, see my detailed comments).

Line 22:  The most optimally oriented fracture sets that require the least injection pressure to be activated are WNW-ESE and N-S. -> is not clear: optimally for what, for slip or to avoid slip of fracture?

Also, the authors specify an injection pressure, but no injection procedure or scope is specified before this point. Injection must be introduced.

Line 25: geothermal: up to line 25, no geothermal topics are specified. I think it must introduced before.

Line 31: Not so clear. Please specify the meaning of energy characteristics.

Figure 1: The dot points are of different sizes, but no legend is given. 

The colour scale of the legend indicates High-temperature energy, but I suggest to use also “geothermal”

Figure 11 indicates the probability slip for each fracture. Is not clear why the authors use a color scale and how to use this chart in order to select an injection pressure. May be necessary to add another chart in which, as a function of the pressure, what is the probability of slipping of the surveyed fractures of the area.

Comments on the Quality of English Language

The English is sufficient, but  a review by a native English speaker is needed

Author Response

Dear reviewer,

Thank you for your thorough reading of the manuscript and comments. The quality of the manuscript was greatly improved. In the next lines, point-by-point explanations are given to address each comment made.

General comments:

The paper is related to the computation of the probability of slip activation in EGS systems using the Montecarlo simulation method.

As general comment, the title doesn’t reflect the content of the paper. I suggest modifying it by adding “enhance geothermal energy”, for example.

The title was modified to “Slip activation potential of fractures in the crystalline basement rocks of Kuujjuaq (Nunavik, Canada) to assess enhanced geothermal systems development”

Also, the abstract doesn’t reflect the rest of the paper. The abstract must be carefully rewritten as it is not clear and does not introduce the current work adequately.

This study aims to estimate the slip activation potential of pre-existing fractures in the crystalline rocks of Kuujjuaq, a community in northern Canada, using outcrop analogues and stress estimates through an empirical approach. A probabilistic slip tendency approach was employed to assess the probability of slip of the pre-existing fractures at the current state of stress, which orientations of fractures are most likely to be activated and what stresses and fluid pressures are needed for the slip activation of pre-existing fractures. This is a key aspect in Enhanced Geothermal Systems.

The results of our analysis suggest that the fractures are relatively stable at the current state of stress, and that the optimally oriented fractures to slip belong to the sets WNW-ESE and N-S. A wellhead injection pressure of about 12.5 MPa/km could be necessary to activate these fractures under empirical conditions.

We believe our abstract reflects the main results of the paper summarize above. Nevertheless, we made few changes to make the abstract clearer.

The results presentation is weak (just Figure 11, which also needs improvement, see my detailed comments).

Line 22:  The most optimally oriented fracture sets that require the least injection pressure to be activated are WNW-ESE and N-S. -> is not clear: optimally for what, for slip or to avoid slip of fracture?

The term activated was used to indicate fractures that will slip. A sentence was added to the abstract to make it clearer.

Also, the authors specify an injection pressure, but no injection procedure or scope is specified before this point. Injection must be introduced.

A sentence was added to the abstract to introduce injection.

Line 25: geothermal: up to line 25, no geothermal topics are specified. I think it must introduced before.

Geothermal was added at the beginning of the abstract to introduce the topic.

Line 31: Not so clear. Please specify the meaning of energy characteristics.

The sentence was modified to “Canada is an energy-intensive and energetically contrasting country”. Furthermore, all this paragraph was re-written to make the message clearer.

Figure 1: The dot points are of different sizes, but no legend is given.

The colour scale of the legend indicates High-temperature energy, but I suggest to use also “geothermal”

The Figure was modified according to the reviewer comments.

Figure 11 indicates the probability slip for each fracture. Is not clear why the authors use a color scale and how to use this chart in order to select an injection pressure. May be necessary to add another chart in which, as a function of the pressure, what is the probability of slipping of the surveyed fractures of the area.

We have updated the figure. The hot color curve represents fractures with a higher slip tendency. As injection pressure increases, the likelihood of fractures slipping also rises. The probability of slip is directly correlated with injection pressure. In the context of our study, if the injection pressure is raised to 26 MPa/km, the probability of slip for the most probable fracture increases to 68%. Further increasing the injection pressure to 30 MPa/km leads to a substantial increase in the probability of slip, reaching 95%.

Round 2

Reviewer 2 Report

Comments and Suggestions for Authors

The authors fully replied to the reviewer's comments and suggestions. I think that now the paper can be published

Comments on the Quality of English Language

The English is sufficient